# MMFSyn: A Multimodal Deep Learning Model for Predicting Anticancer Synergistic Drug Combination Effect

**DOI:** 10.3390/biom14081039

**Published:** 2024-08-22

**Authors:** Tao Yang, Haohao Li, Yanlei Kang, Zhong Li

**Affiliations:** 1School of Information Engineering, Huzhou University, Huzhou 313000, China; 202120102065@mails.zstu.edu.cn; 2College of Science, Zhejiang Sci-Tech University, Hangzhou 310018, China; hhl820@zstu.edu.cn

**Keywords:** synergistic drug combination, multimodal data, SMILES, deep learning

## Abstract

Combination therapy aims to synergistically enhance efficacy or reduce toxic side effects and has widely been used in clinical practice. However, with the rapid increase in the types of drug combinations, identifying the synergistic relationships between drugs remains a highly challenging task. This paper proposes a novel deep learning model MMFSyn based on multimodal drug data combined with cell line features. Firstly, to ensure the full expression of drug molecular features, multiple modalities of drugs, including Morgan fingerprints, atom sequences, molecular diagrams, and atomic point cloud data, are extracted using SMILES. Secondly, for different modal data, a Bi-LSTM, gMLP, multi-head attention mechanism, and multi-scale GCNs are comprehensively applied to extract the drug feature. Then, it selects appropriate omics features from gene expression and mutation omics data of cancer cell lines to construct cancer cell line features. Finally, these features are combined to predict the synergistic anti-cancer drug combination effect. The experimental results verify that MMFSyn has significant advantages in performance compared to other popular methods, with a root mean square error of 13.33 and a Pearson correlation coefficient of 0.81, which indicates that MMFSyn can better capture the complex relationship between multimodal drug combinations and omics data, thereby improving the synergistic drug combination prediction.

## 1. Introduction

Cancer, cardiovascular disease, and many other diseases exhibit heterogeneity with different pathological features and epigenetic variations, resulting in different responses and resistances to drugs. Additionally, due to their biological complexity involving multiple target genes, single-drug therapy often cannot fully perform well [1,2,3,4]. Therefore, both traditional medicine and modern medicine have recently utilized the advantages of using multiple drugs in combination to treat diseases. Combination therapy can act on different molecular targets of tissue cells, thereby improving efficacy, reducing side effects [5], and overcoming drug resistance [6]. The comprehensive effect brought by this combination of drugs, which is called drug synergy, often exceeds the total effect of using each drug alone. Drug combination therapy has increasingly been used to treat various complex diseases, such as hypertension, infectious diseases, and cancer, by simultaneously acting on different targets or biological processes [7]. Therefore, it is crucial to accurately identify synergistic drug combinations targeting specific diseases.

At present, the main methods for predicting drug combination effects include biological experiment-based methods, machine learning-based methods, and deep learning-based methods. Traditional discovery of drug combinations is mainly based on clinical biological trials, limited to a small number of drugs [8], which is far from meeting the urgent demand for anti-cancer drugs. With the development of high-throughput drug screening technology, researchers can simultaneously conduct large-scale drug combination screening on hundreds of cancer cell lines. For example, Torres et al. [9] used yeast to screen a large number of drug combinations and provided a method for identifying preferred drug combinations for further testing in human cells. In order to screen effective antimicrobial drug combinations for in vivo disease treatment, technologies such as microcalorimetry screening and genetically encoded fluorescent biosensors have been developed [10]. However, these technologies require proficient operation and complex experimental procedures, which have extremely high requirements for relevant practitioners. Given the high cost and unsatisfactory efficiency of pharmaceutical research and clinical trials due to the large number of drug combinations, identifying synergistic effects between drugs remains a challenging task [11].

With the development of computer science and biomedical fields, it has become possible to use computational methods to predict the effectiveness of drug combinations, as well as drug–drug interactions, which is another way to identify drug synergy [3,4,12,13,14,15,16]. These computational methods can help researchers quickly and accurately evaluate the potential effects of different drug combinations, thereby accelerating the process of drug development. Researchers can also apply machine learning methods to explore drug combination associations for determining synergistic therapeutic effects. For example, Li et al. [17] applied support vector regression (SVR) to successfully predict that a new derivative of dihydrofuran-2-one (LPP1) combined with pregabalin would have the greatest analgesic effect in a streptozotocin-induced neuropathic pain model in mice. Liu et al. [18] trained a gradient-enhanced tree classifier by running random walks with restart (RWR) on a drug–protein heterogeneous network and used these features to predict new drug combinations.

Recently, deep learning has increasingly been applied in the development and discovery of multiple drugs. For example, the DeepSynergy model combines chemical information of drugs with the genomic data of cancer cells to predict drug combinations with synergistic effects [19]. TranSynergy [20] is a self-attention mechanism-based deep learning model that integrates information from gene–gene interaction networks, gene dependencies, and drug target associations to predict synergistic drug combinations, aiming to reveal the synergistic mechanisms and pathways of drug combinations. The MatchMaker model [21] chooses the chemical structure information of drugs and gene expression data of cell lines as inputs to predict synergistic drug combinations and preliminarily explain their molecular mechanisms. The DTSyn model [22] is based on a multi-head attention mechanism to identify new drug combinations. It designs a fine-grained transformer to capture chemical substructure–gene and gene–gene associations, as well as a coarse granularity transformer to extract chemical–chemical and chemical–cell line interactions, integrate drug and cell line information, and, finally, achieve the prediction of synergistic effects. On the other hand, some studies have applied simplified molecular input line entry system (SMILES) encoding to characterize the chemical properties of drugs. For example, Sun et al. [23] proposed a deep tensor factorization model that combines the tensor factorization with a feedforward neural network (FNN) for predicting drug synergistic effects. In addition, researchers also used graph neural networks to learn feature representations of drug chemical structures. For example, Wang et al. [24] proposed a deep learning model called DeepDDS based on graph neural networks and attention mechanisms, which embeds drug molecular structure and gene expression data features as inputs into FNN to identify drug combinations, which can effectively inhibit specific cancer cell viability.

At present, there are still challenges in predicting the combined effects of anti-cancer drugs. Firstly, most existing methods rely on manually extracted features from professional wet experiments, which limits the scope of application of these methods. Secondly, existing methods mainly focus on feature processing of drugs based on their chemical substructures, drug target correlation information, etc. Compared to these data, drug SMILES encoding is easier to obtain and contains multiple modal data. How to deeply mine information from SMILES and construct a multimodal feature training network for predicting still remains a challenge. This is the main content that will be addressed by our proposed MMFSyn.

## 2. Materials and Methods

### 2.1. Synergy Dataset

#### 2.1.1. Benchmark Dataset

To objectively compare our model with existing methods, we use the common dataset, namely, the large-scale collaborative dataset published by O’Neil et al. [25], as the benchmark dataset for the prediction model training and evaluation. This dataset covers a total of 23,062 pieces of data from 39 cancer cell lines in seven tissues with 583 different drug combinations.

There are 38 drugs in this dataset, consisting of 24 FDA-approved drugs and 14 experimental drugs. In order to evaluate the synergistic effect of these drugs on specific cancer cell lines, we adopt the method of Preuer et al. [19] and integrate the dataset into the synergistic effect score between two drugs and a certain cancer cell line by calculating the Loewe additivity value [26]. In terms of data format, this dataset contains 23,062 quadruples, each including two drugs, a cancer cell line, and corresponding synergistic effect scores. For the model training and evaluation, we divide them into the training set and the test set at a ratio of 9:1. A 5-fold cross-validation is performed based on these data, which includes the training and validation process.

#### 2.1.2. Independent Test Dataset

In order to validate the performance of our model on a completely new test set and evaluate the synergistic interaction of drug combinations, this study collects drug combination data from the AstraZeneca–Sanger Drug Combination Prediction DREAM Challenge [27]. This challenge compiles a combination drug sensitivity screening dataset, encompassing 11,576 experimental data. We compare the AstraZeneca–DREAM (AZ–DREAM) challenge dataset with prior drug combination data, discovering overlaps in 10 cancer cell lines, seven related targets, and four drugs, but no drug combination–cell pair repetitions. After the removal of duplicate data, the drugs and cancer cell lines are screened, excluding drug molecules not found in Drugbank and cancer cell line expression data not found in the CCLE database. Ultimately, an independent test set is constructed, which includes 668 drug pair–cell line combinations, covering 57 drugs and 24 cancer cell lines.

### 2.2. MMFSyn Model

The main framework of our MMFSyn model is shown in Figure 1, which utilizes four drug features: Morgan fingerprint, representing a one-dimensional structure [28]; an atom sequence and molecular diagram, representing two-dimensional information; and an atomic 3D point cloud data to construct modules for feature extraction of drug combinations in different modalities. The atom sequence is usually used to represent the structure and composition of molecules; the molecular diagram provides a more detailed description of molecular connection and topological structure, while the 3D point cloud of atoms provides information about the atomic position and molecular conformation. The utilization of these data can provide a more comprehensive understanding of the drug characteristics.

Specifically, we first utilize RDKit to extract the one-dimensional Morgan fingerprint features of drugs. Secondly, the BiLSTM-gMLP network is used to extract single-drug sequence features, and then the multi-head attention mechanism is applied to fuse and reduce the dimensionality of two drug sequence features to obtain drug combination sequence features. Thirdly, in the feature extraction of molecular diagram, for individual drug and drug combinations, feature extraction and fusion are performed using graph convolutional neural network modules at three scales: the first-order power graph, the second-order power graph, and the third-order power graph, respectively, to obtain molecular diagram features of drug combinations. These features are then integrated with drug sequence features to obtain two-dimensional drug combination features. Fourthly, another feature extraction is performed on drug atomic point cloud information through one-dimensional convolution and residual neural networks to obtain the three-dimensional features of drug combinations. Fifthly, we use the feedforward neural network (FNN) to combine these modules for fusing and learning the multimodal features of drug combinations. Additionally, our model applies FNN for dimensionality reduction and fusion of gene expression and mutation data to obtain the characteristics of cancer cell lines. Finally, these multimodal features and multi-omics cancer cell line features of drug combinations are input into a predictor for predicting the synergistic anti-cancer drug combination.

### 2.3. Drug Features

#### 2.3.1. One-Dimensional Feature Processing of Drugs

In order to represent the one-dimensional structural features of drugs, we use the Morgan fingerprint to construct the feature vector for each drug. The SMILES expression of each drug contained in the O’Neil dataset forms a specification for drugs to clearly describe their molecular structure using ASCII strings. Here, we apply RDKit to calculate the Morgan fingerprint of each drug based on the drug SMILES expression. The Morgan fingerprint algorithm is a commonly used molecular fingerprint representation method [28], which can be used to describe the structural information of molecules. Morgan fingerprints are based on the connectivity of molecules, recording the environmental information around each atom in the molecule. Here, we generate Morgan fingerprints with a radius of 2 for each drug and represent them as 256-dimensional binary value vectors, i.e.,:$$Morgan_{i}=\mathrm{fingerprint}(SMILES_{i})$$
where $Morgan_{i}$ is the Morgan fingerprint feature of the *i*-th drug, and fingerprint(·) is the operation using RDKit to process the Morgan fingerprint. For the drug combination, its Morgan fingerprint feature is represented as $H_{m}=Morgan_{1}$ || $Morgan_{2}$. Here, “||” represents the splicing operation.

#### 2.3.2. Sequence Feature Processing of Drugs

For the extraction of two-dimensional sequence information of drugs, we first extract the atomic features in the drug sequence from SMILES. Here, a set of atomic features adapted from DeepChem is used for this purpose [29]. We process the extracted raw features to obtain drug sequence encoding data $X\in R^{N\times C}$ (where *N* is the number of atoms and *C* is the number of features per atom), then, we input *X* into a BiLSTM layer that captures the interdependence between drug sequence atoms and obtain an output representation $h_{t}\in R^{2d_{l}}$, where *d_l_* represents the number of output units used in each LSTM unit, namely, $h_{t}=\vec{\mathrm{LSTM}}\left(X_{t},h_{t-1}\right)||\overset{\leftarrow }{\mathrm{LSTM}}\left(X_{t},h_{t-1}\right)$.

Secondly, considering that BiLSTM extracts features from both forward and backward directions, we use the gMLP module to capture the features extracted from the forward direction and perform the dimensionality reduction and quadratic encoding to obtain the features of a single drug. The calculation formula for a single gMLP module is:
*Z* = *σ*(*h_t_U*), *Y* = *s*(*Z*)*V*, *s*(*Z*) = *Z*_1_(*WZ*_2_ + *b*)
where *σ* is the activation function, *U* and *V* represent the trainable linear mapping parameter matrix, *W* is the spatial interaction mapping parameter matrix, *s*(.) is the spatial gating unit (SGU), and *Z*_1_ and *Z*_2_ are divided into two parts by *Z* from the channel dimension for the gate operation. Here, the feature *H_s_* of a single drug is obtained by repassing *h_t_* from BiLSTM to multiple layers of gMLP.

Finally, two drug sequence features *H_s_*_1_ and *H_s_*_2_ are concatenated element-by-element to obtain a vector, which is then fed into an encoder with a multi-head self-attention mechanism for feature dimensionality reduction and fusion. Namely, we first define the input representation as $X_{c}=\left[H_{s1}⊕H_{s2}\right]\in R^{N_{t}\times d_{t}}$, where *N_t_* is the sequence length of *H_s_*_1_; *H_s_*_2_, *d_t_* is the input dimension representing the output features through the BiLSTM-gMLP network; and ⊕ represents the point-by-point addition operation. Then, $X_{c}$ is passed through *h* different linear mappings to obtain *Q_i_* = *X_c_W_Qi_*, *K_i_* = *X_c_W_Ki_* and *V_i_* = *X_c_W_Vi_*, where the dimensions of *W_Qi_*, *W_Ki_*, and *W_Vi_* are *d* × *d_q_*, *d* × *d_k_* and *d* × *d_v_*, respectively. We calculate the attention scores for each head:$$\mathrm{Attention}(Q_{i},K_{i},V_{i})=\mathrm{Softmax}(\frac{Q_{i}K_{i}^{T}}{\sqrt{d_{k}}})\;V_{i}$$
where $Q_{i},\;K_{i}$, and $V_{i}$ are the query, key, and value of the *i*-th head. Then, we concatenate the attention values of all heads:
*H_c_* = Concat(Attention_1_, Attention_2_, …, Attention_h_)*W^o^*
where $W^{o}\in R^{\left(d_{v}\times h\right)\times d}$ is the output linear mapping matrix, which is used for obtaining the two-dimensional sequence features of drug combinations.

#### 2.3.3. Molecular Diagram Feature Processing of Drugs

We first apply SMILES to convert each drug into a molecular diagram, whose vertices are atoms, edges are chemical bonds, and vertex features in the molecular diagram are consistent with atom sequence features. Usually, the adjacency representation of any diagram maintains the connectivity relationship between its nodes. Therefore, we understand that the interaction between each atom node and its neighboring nodes is crucial in a molecular diagram. In order to characterize the two-dimensional molecular diagram information of drugs, it is necessary to deeply mine the connectivity relationship between nodes in a graph. Here, we use graph convolutional networks (GCN) to extract the feature from different scales of node connectivity in the molecular diagram.

For a given molecular graph *A*, each atom node *v* can be connected to each node *u* belonging to its neighborhood *Γ*(*v*) through an edge, and we define their distance as 1 jump. For the node *w*, which is in *Γ*(*u*) but not in *Γ*(*v*), we define the distance between *v* and *w* as 2 jumps. In other words, they are two jumps apart. If a node is connected to all such nodes with two jumps, we can obtain a new second-order power graph *A*^2^. Similarly, in order to further express the local adjacency, a third-order power graph *A*^3^ can also be constructed.

In the feature processing network of two-dimensional molecular diagrams, we first introduce the power graph module composed of GCN, which is divided into three parts: the first part stacks three GCN layers, the second part stacks two GCN layers, and the last section uses a GCN layer. For each drug, the adjacency matrix *A*$\in R^{N\times N}\;$ and node feature matrix $X\in R^{N\times C}$ are used as inputs for the first block, and GCN is used for message passing in the drug molecular diagram, as shown in Figure 2a.

For the first part of the power graph module by GCN, in order to overcome the degree normalization problem of adjacency representation, we calculate the normalized adjacency representation as:$$A_{norm}=D^{-1/2}AD^{-1/2}$$
where $D\in R^{N\times N}$ is the degree matrix of *A*. Then, we use the following process
$$H_{e}^{i}=\sigma (A_{norm}H_{e}^{\left(i-1\right)}W^{\left(i-1\right)})$$
to obtain the feature of the drug compound produced in the *i*-th layer regarding *A*, where *W*^(*i*−1)^ is a trainable parameter, and $H_{e}^{0}=X$ is the output by the *i*-th layer.

In the second part of the GCN power graph module, a second-order power graph $A^{2}\in R^{N\times N}$ and the same node features *X* are used as inputs. Similar to the calculation method of the first-order power graph, the normalized adjacency representation is first calculated by $A_{norm}^{2}=D^{\prime -1/2}A^{2}D^{\prime -1/2}$, then, we obtain the feature of drug produced in the *i*-th layer regarding *A*^2^ by $H\prime_{e}^{i}=\sigma \left(A_{norm}^{2}H\prime_{e}^{\left(i-1\right)}W\prime^{\left(i-1\right)}\right)$, where $W^{\prime \left(i-1\right)}$ is a trainable parameter and $H\prime_{e}^{0}=X$ is the output by the *i*-th layer.

Similarly, we operate $A_{norm}^{3}=D^{\prime \prime -1/2}A^{3}D^{\prime \prime -1/2}$ and ${H^{\prime \prime }}_{e}^{i}=\sigma \left(A_{norm}^{3}{H^{\prime \prime }}_{e}^{\left(i-1\right)}W^{\prime \prime \left(i-1\right)}\right)$ to obtain the feature of drug produced in the *i*-th layer regarding *A*^3^. Finally, we concatenate these output representations of three parts and use a global max pooling (GMP) function suitable for the graph structure for fusion and dimensionality reduction to obtain the final two-dimensional molecular diagram representation of each drug.
$$H_{e}=\mathrm{GMP}(H_{e}^{i}\left|\left|{H^{\prime }}_{e}^{i}\right|\right|{H^{\prime \prime }}_{e}^{i})$$

Considering that the problem addressed in the prediction model is the synergistic effect of two drugs, for drug one and drug two, their adjacency matrix *A_d_*_1_, *A_d_*_2_ and node feature matrix *X_d_*_1_, *X_d_*_2_ are obtained, respectively, and their corresponding concatenation is used to obtain the adjacency matrix *A_dc_* and node feature matrix *X_dc_* of the drug combination. Then, the corresponding data of drug one, drug two, and drug combination are fused and embedded through the molecular diagram feature fusion module for information transmission, which is characterized separately by *H_e_*_1_, *H_e_*_2_ and *H_ec_*, respectively, and finally concatenated to obtain the overall two-dimensional molecular diagram representation of two drug combinations, as shown in Figure 2; namely:$$H=H_{e1}||H_{ec}||H_{e2}$$

#### 2.3.4. Three-Dimensional Atomic Point Cloud Feature Processing of Drugs

When using RDKit to process drug SMILES, relevant information including atomic coordinates is also collected. We construct a point cloud feature embedding network after obtaining the 3D atomic coordinates of drugs, as shown in Figure 3.

Firstly, the 3D atomic point coordinates of each drug in the drug pair are processed as *P_D_* = {*p_d_*_1_, *p_d_*_2_, …, *p_dm_*} based on SMILES, where *p_di_* represents a 3D atomic point in the drug molecule and *m* is the total number of atomic points. Considering the different lengths of each drug sequence, we perform the data completion to obtain the coordinates of each drug with the standard length, whose shape is *n* × 3 (*n* is the processed standard drug sequence length). Therefore, the 3D atomic point features of each drug are represented as *P_D_* = {*p_d_*_1_, *p_d_*_2_, …, *p_dn_*}.

Secondly, two drugs in the drug combination are embedded into the network through atomic point cloud features, and after multi-layer one-dimensional convolution, BatchNorm1d and ReLU activation function with residual connection, the atomic point cloud features of each drug become *H_D_*, which is represented as:
*H_D_* = Residual(…Conv1D(Residual(…Conv1D(*P_D_*)…))…)
where Conv1D( ) represents one-dimensional convolution operation and Residual( ) denotes the residual connection (BatchNorm1d and activation function are omitted in the formula).

Specifically, for the atomic point cloud data input of each drug with *n* × 3, the embedding network maps it to a high-dimensional 32-dimensional space by combining one-dimensional convolution and residual structure, then aligns it in a 64-dimensional space and a 128-dimensional space, respectively, and finally maps it to a 160-dimensional space to form the feature vector.

### 2.4. Cell Line Features

For the feature extraction from cancer cell lines, we integrate two types of cell line data, namely gene expression data and gene mutation data, accompanying tissue information, to construct the features of cell lines.

Gene expression data are downloaded from the ArrayExpress database (login number: E-MTAB-3610) [30]. A total of 3384 informative genes are first summarized using factor analysis for robust microarray summarization (FARMS) [31]. FARMS is a factor analysis method used for microarray data, aimed at extracting signals from noisy microarray data. It achieves this goal by representing gene expression data as a linear combination of potential factors. Compared with general microarray data analysis methods, FARMS is more robust, capable of handling noise and outliers and providing more accurate and robust gene expression estimates. After using FARMS to reduce the dimensionality of gene expression data, the final result is described as a gene load matrix *G* with a size of *p* × *k*, where *p* is the number of cell lines and *k* is the specified number of potential factors. This matrix contains the weights of each gene on each potential factor, reflecting the expression patterns of genes in different biological processes. Then, the normalization operation is performed through the *z*-score, and the standardized gene load matrix is expressed as *G*′.

The gene mutation data of cell lines come from the COSMIC cell line project [32]. We remove those data with coding silent or unknown mutation types and retain mutation data for 10,707 genes from 39 cell lines, namely, the gene mutation data for each cell line is represented as a 10,707-dimensional binary value vector. According to whether the cell line undergoes genetic mutations, the corresponding element of the vector is either 0 or 1. The final gene mutation data matrix is obtained and denoted by *M*.

For two omics cell line data, we combine them and use the feedforward neural network (FFN) for the dimensionality reduction to obtain the fusion features of multiple omics cell lines, which can be used in the subsequent prediction task along with extracted drug features.
$$F_{cell}=\mathrm{FNN}(G^{\prime }||M)$$

### 2.5. Predicting Module

After constructing multimodal features and multi-omics cancer cell line features of drug combinations, we design a predicting module to predict the synergistic score of drug combinations with cell lines. This module receives fusion features *H_m_* || *H_e_* || *H_g_* || *H_D_* of drug combinations and cell line feature *F_cell_* as inputs, which is set by three fully connected layers. Among them, the first two fully connected layers use the ReLU activation function, followed closely by the batch normalization layer. In addition, the number of neurons in the second fully connected layer is set to half of that in the first fully connected layer, and the last fully connected layer only contains one neuron, which represents the collaborative score predicted by the model.

Considering that predicting the combination of anticancer drugs is a regression task whose prediction result is the synergistic score of the combination of drugs against cancer. In our model, the loss function chosen for training is set by the mean square error loss:$$Loss=\frac{1}{\left|T\right|}\sum_{(d,d,c)\in T}{\left(P_{ddc}-T_{ddc}\right)}^{2}$$
where *T* is the training set of drug–drug–cell line (*d*, *d*, *c*), *T_ddc_* represents the true quantitative collaborative score, and *P_ddc_* represents the model prediction result.

## 3. Experimental Setup

### 3.1. Evaluating Indicator

For the collaborative prediction task of anti-cancer drugs, we select four main indicators to evaluate the performance of the model, namely, mean square error (MSE), root mean square error (RMSE), the Pearson correlation coefficient (PCC), and the Spearman correlation coefficient (SCC). These indicators comprehensively consider the performance of the model in data fitting, linear correlation, prediction accuracy, and interpretability ability.

### 3.2. Model Settings

During our model training, the batch size in our network training is set to 256, AdamW is used as the optimizer of the model, and the learning rate change strategy is StepLR. The main process of StepLR is as follows: firstly, we initialize the learning rate and attenuation factor; secondly, at the end of each epoch with specified step_size, we update the learning rate and set it to be the current learning rate multiplied by the decay factor; finally, we repeat the previous step until the set training round is reached. The StepLR strategy is simple and easy to use, which can effectively control changes in the learning rate, accelerate the model convergence, and improve the predicting performance. In our model, we set the initial learning rate to 0.001, the gamma parameter to 0.9, and the step_size to 20, which means that for every 20 epochs, the learning rate will decay at a ratio of 0.9.

Our proposed multimodal deep learning architecture requires substantial computational resources for implementation in training. To expedite the training process, we can preconfigure the dataset, allowing us to maximize the computational efficiency. The detailed implementation can be observed in our GitHub repository, demonstrating how we explore the full potential of computing resources.

## 4. Results

### 4.1. Detailed Prediction Result Analysis

For the large-scale collaborative dataset based on O’Neil et al. [25], we train and predict the collaborative drug combination effect using our MMFSyn model and present a density plot of the predicted coordination score and the actual coordination score, as shown in Figure 4a. Through the presentation of density maps, it can be visually observed that the data distribution between the predicted results of the model and the true values is similar. Figure 4b shows the correlation between collaborative scores predicted by the MMFSyn model and the actual score, and the red line fitted by the least square method shows the functional relationship between the predicted score and the actual situation. By computation, we find its slope is 0.907 and its bias is 0.584, indicating a significant linear correlation between the model’s predicted results and the actual situation. Namely, the overall prediction deviation of the model is small, and the predicted results have high accuracy and credibility. Figure 4c shows the experimental results of 5-fold cross-validation based on the MMFSyn model using a box plot, with an MSE of 178.13 and a PCC of 0.81. This indicates a strong linear correlation between the predicted results of the model and the actual observed values, further verifying the reliability and effectiveness of the model.

In addition, in order to further explore the predictive results of the model in different tissue types, we use a double-sided violin plot to display the distribution of true collaborative scores of cell lines derived from each tissue and the distribution of predicted collaborative scores by the model, which is shown in Figure 5a. This study finds that in skin, ovarian, lung, colon, and breast tissue, the distribution of true collaborative scores and predicted collaborative scores is relatively stable, concentrated in the range of [−50, 75]. However, in contrast, in pleura, the median, mean, quartile, and other predicted collaborative scores are higher than the true collaborative scores. In pleura, the difference between the predicted collaboration score and the actual collaboration score is relatively significant. This difference may be related to the small number of cell lines belonging to the given organization in the dataset and the dispersed distribution of collaborative scores.

Furthermore, we provide the predicted visualization results of different organizational types. A box plot of PCC values classified by organization is shown in Figure 5b and a box plot of MSE values classified by organization is shown in Figure 5c. In these graphs, the middle line of each box represents the median. On the 5-fold cross-validation test set, it is found that the average PCC values of different tissue types show a certain difference; for example, pleura is 0.68, prostate is 0.79, ovarian is 0.80, colon is 0.81, and skin is 0.82, while breast and lung are 0.83. The box plot of MSE values in Figure 5c reflects the prediction error of different organizational types, and there exists also a certain difference in the average MSE values of different organizational types. Specifically, the colon is 110.72, the ovary is 159.89, the skin is 154.05, the lung is 197.53, the breast is 165.16, the prostate is 380.96, and the pleura is 651.03.

We also conducted an analysis of the results according to different cell lines. Figure 6 shows the average PCC value and average MSE value in the 5-fold cross-validation of each cell line, and the color of the bar graph represents the tissue to which the cell line belongs. Among thirty-nine cell lines, only one cell line has a PCC value below 0.6, while twenty-seven cell lines have a PCC value above 0.8. The cell lines with the highest and lowest PCC values belong to the ovarian tissue. Among them, the PCC value of A2780 cell line is the highest, with 0.89; the PCC value of UWB1289BRCA1 cell line is the smallest, with 0.58. In addition, we find only three cell lines have MSE values above 500, and eleven cell lines have MSE values below 100. The MDAMB436 cell line belonging to breast tissue has the lowest MSE value, with 46.75. The MSTO cell line belonging to pleura has the highest MSE value, with 643.28.

### 4.2. Ablation Study

#### 4.2.1. Ablation Study for Drug Features

We use the ablation experiment to analyze the impact of each component in MMFSyn based on the 5-fold cross-validation results. Figure 7 shows the performance of different variants of MMFSyn on the dataset. Firstly, we set the baseline model MMFSyn_base, which includes drug Morgan fingerprint features and 3D atomic point cloud features. Secondly, we introduce the drug sequence feature and fuse drug features based on whether a multi-head attention mechanism encoder is used or not, and the models are named Seq_A and Seq_B, respectively. Here, Seq_A uses a multi-head attention mechanism encoder, and Seq_B does not use it. Thirdly, we introduce drug molecular graph features, which are divided into Graph-A and Graph-B based on whether the molecular graph feature fusion embedding module is used or not. Here, Graph-A means that the molecular graph feature is used and Graph-B is not used. In addition, an MMFSyn-add model is also set up, which uses all feature processing modules in MMFSyn and applies the self-attention mechanism to fuse multimodal features.

From the results of the 5-fold cross-validation, it can be observed that the model with adding drug sequence features (Seq_A) and the model with adding drug molecular map features (Graph_A) are both better than the baseline method using only the Morgan fingerprint and 3D atomic point cloud (MMFSyn_base), where MSE values are reduced by 17.17 and 18.98, respectively. Within the model for processing drug sequence features, the prediction result of Seq-A using a multi-head attention mechanism encoder is decreased by 4.55 compared to Seq-B not using it; Within the model that processes molecular graph features, Graph_A using the fusion embedding module has an MSE value that is 1.77 lower than Graph_B, which is not used.

Moreover, considering that the PERCEIVER IO method proposed by Jaegle et al. [33] extended the cross attention to multimodality and achieved good results in related tasks, we also analyze the use of cross attention mechanism and self-attention mechanism to fuse multimodal drug combination features and cancer cell line multi-omics features as variants MMFSyn-CA and MMFSyn-SA. Compared to the MMFSyn method, the MSE values of MMFSyn-CA and MMFSyn-SA are higher, with 7.65 and 10.89, respectively, indicating that the effect of MMFSyn-CA and MMFSyn-SA is worse than that of MMFSyn. Therefore, the cross-attention mechanism or self-attention mechanism is not used for fusion. Based on the comparison and analysis of the above results, it can be concluded that the various feature extraction modules and predictor settings used by MMFSyn have indeed improved the performance for predicting the combination of anticancer synergistic drugs, namely, all model parts are indispensable.

#### 4.2.2. Ablation Study for Cell Lines

We also conduct the ablation experiment on cell lines, analyzing the experimental effect of using gene expression profiles and mutation data separately. Firstly, we establish a baseline model Cell_base, which neither uses gene expression data nor mutation data but instead employs one-hot encoding to differentiate cancer cell lines. Secondly, we set up the model Cell_gen, which only uses gene expression data, keeping the drug feature processing part unchanged. Thirdly, we introduce mutation data features only by setting up the model Cell_mut, again keeping the drug feature processing part unchanged. We compare these three models to the overall MMFSyn, and the results are listed in Table 1.

As shown in Table 1, MMFSyn outperforms other models in terms of MSE. Specifically, the average MSE of the Cell_base model is 214.36, while the average MSEs of Cell_gen and Cell_mut are 185.27 and 189.69, respectively. This indicates that the use of gene expression data alone yields superior results compared to using mutation data alone. Moreover, the MMFSyn model, which integrates both types of data, demonstrates the best performance (the average MSE is 178.13), highlighting the synergistic effect of combining gene expression and mutation data.

### 4.3. Different Prediction Method Comparison

To evaluate the effectiveness of the proposed MMFSyn, we conduct comparative experiments with various existing methods, including machine learning methods (RF [34], XGBoost [35], Linear Regression, and Elastic Net [36]), as well as the deep learning methods (PRODeepSyn [37], TranSynergy [20], AudnnSynergy [38], DeepSynergy [19]). All these methods are specifically designed for drug combination synergy prediction and have been tested with their optimal parameters as described in their respective papers. PRODeepSyn integrates protein–protein interaction network information into omics data and utilizes GCN to construct cell line characterization, providing an efficient computational model for discovering new synergistic anticancer drug combinations. The design of TranSynergy enables clear modeling of the cellular effects of drug action through cell line gene dependence, gene–gene interactions, and genome-wide drug target interactions. AuDNNsynergy trains three autoencoders using gene expression, copy number, and gene mutation data from tumor samples from the Cancer Genome Atlas (TCGA) [39], integrating multiple omics data to predict the synergistic effects of paired drug combinations. For a more effective comparison, we implement the 5-fold cross-validation on the dataset. The detailed comparison results between MMFSyn and these advanced methods are shown in Table 2.

As shown in Table 2, bold values indicate the highest performance in each category. MMFSyn performs better than other methods in MSE, RMSE, PCC, and SCC. Specifically, the values of these four indicators are 178.13, 13.33, 0.81, and 0.80, respectively. Compared with these popular methods, the MMFSyn method has improved by 30.36 on the MSE indicator, 1.21 on the RMSE indicator, 0.06 on the PCC indicator, and 0.06 on the SCC indicator.

### 4.4. Evaluate the Model in an Independent Test Set

In order to better verify the generalization ability of our model, the benchmark dataset (i.e., the original training set) is used to train the model, and then an independent test set released by AstraZeneca is used to evaluate the performance of our prediction model MMFSyn and other competitive methods. This independent test set includes 668 drug pair–cancer cell line combinations, covering 57 drugs and 24 cell lines. The prediction results of MMFSyn and comparative methods on this independent test set are shown in Table 3. It can be seen that the performance of MMFSyn is superior to all other comparative methods in the main performance indicator of MSE.

### 4.5. Case Study

We conduct the case study analysis using the prediction results of MMFSyn and find that many cases are consistent with previous studies. For example, Gil-Martin et al. [40] tested the therapeutic effect of BEZ-235 and Paclitaxel on breast cancer patients. The experiment did not obtain any evidence that this drug combination had a synergistic effect, and the subjects suffered from various adverse reactions. The prediction result given by MMFSyn is consistent with this experiment. On breast cancer cell lines OCUBM and EFM192B, the synergy scores predicted by MMFSyn are −5.39 and −16.69, respectively. In addition, Wisinski et al. [41] confirmed that the combination of MK-2206 and Lapatinib could be tolerated with a higher dose than monotherapy. They conducted in vitro experiments on HCT-15 to evaluate the mechanism of this drug interaction. MMFSyn gives higher predicted synergy scores on DLD1, HT29, and LOVO cell lines, which are used to study the same types of cancer as the HCT-15 cell line, with 49.69, 35.29, and 42.43, respectively. Furthermore, Lara et al. [42] argued that the therapeutic effect of the combination of MK-2206 and Erlotinib on patients with non-small cell lung cancer (NSCLC) is worthy of further exploration. We check the prediction results of MMFSyn for three NSCLC cell lines included in the dataset, namely SKMES1, NCIH460, and NCIH520, which are 42.36, 18.96, and 21.37, respectively. These results also indicate that the combination of MK-2206 and Erlotinib may show a synergistic effect in the treatment of NSCLC.

## 5. Discussion and Conclusions

This paper proposes an end-to-end deep learning model, MMFSyn, which extracts and fuses features from multimodal drug data and multi-omics data of cancer cell lines. Firstly, it uses the commonly used representation of drugs, SMILES, to comprehensively extract corresponding drug features from different modalities. Next, for different modalities of data, this model uses deep learning modules such as gated multilayer perceptron, graph convolutional neural network, and multi-head attention mechanism to deeply extract features and explore potential information in data. Furthermore, it analyzes the different omics information of cancer cell lines, screens suitable omics features, and fuses them to obtain the fusion feature of cancer cell lines. Finally, these features are combined as multimodal drug features and used to achieve the accurate prediction of synergistic anti-cancer drug combinations. Various experiment analyses and comparisons verify that our MMFSyn obtains satisfactory performance and outperforms other popular prediction methods.

Of course, there is still some work to be conducted to improve the existing methods in the future. Although MMFSyn has integrated various drug data with one-dimensional structure, atom sequence information, molecular diagrams, and three-dimensional atomic point clouds of drugs, there are still other types of drug feature information in practical applications, such as physical properties and biological activities of drugs. We will further explore these features and attempt to integrate them into the model to improve the prediction accuracy and reliability. In addition, we find MMFSyn provides a relatively conservative prediction result for drug combinations that should have high synergistic scores, which may be due to the concentration of synergistic scores near 0 in the training set. With the release of more experimental data, this issue is expected to be further explored and resolved. Finally, we will collaborate with the medicine school or hospital to further perform the web-lab experiment (test on primary patient samples or in vivo models) and incorporate the pathway or gene regulatory network analysis to enhance the biological interpretability.

## Figures and Tables

**Figure 1 biomolecules-14-01039-f001:**
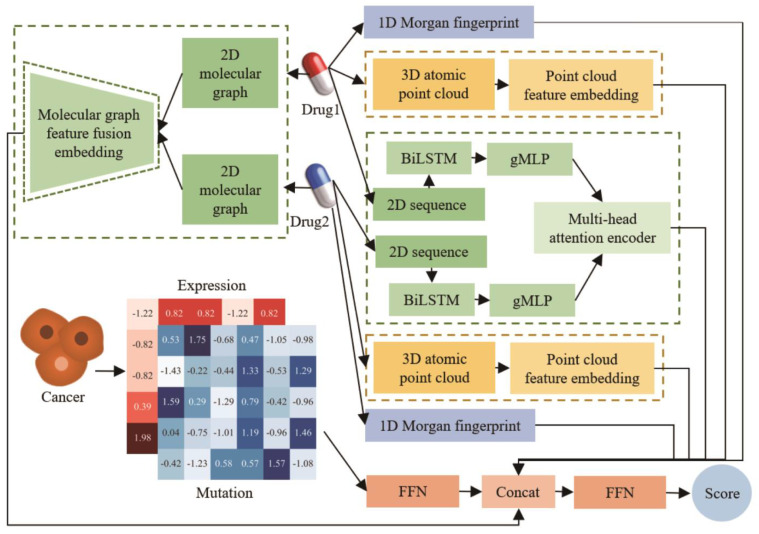
Framework of our anticancer collaborative drug combination prediction model based on multimodal deep learning.

**Figure 2 biomolecules-14-01039-f002:**
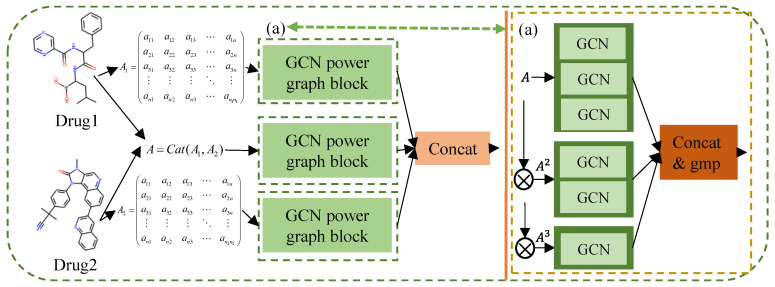
Molecular graph feature fusion embedding process. (a) is the GCN power graph block, and its detailed structure is shown on the right side of the figure.

**Figure 3 biomolecules-14-01039-f003:**
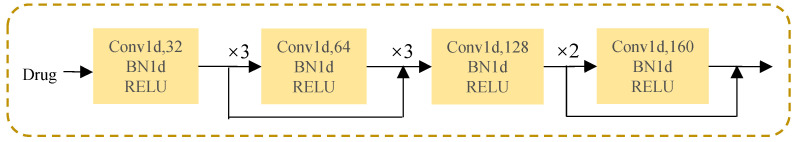
Embedding network for atomic point cloud feature of each drug.

**Figure 4 biomolecules-14-01039-f004:**
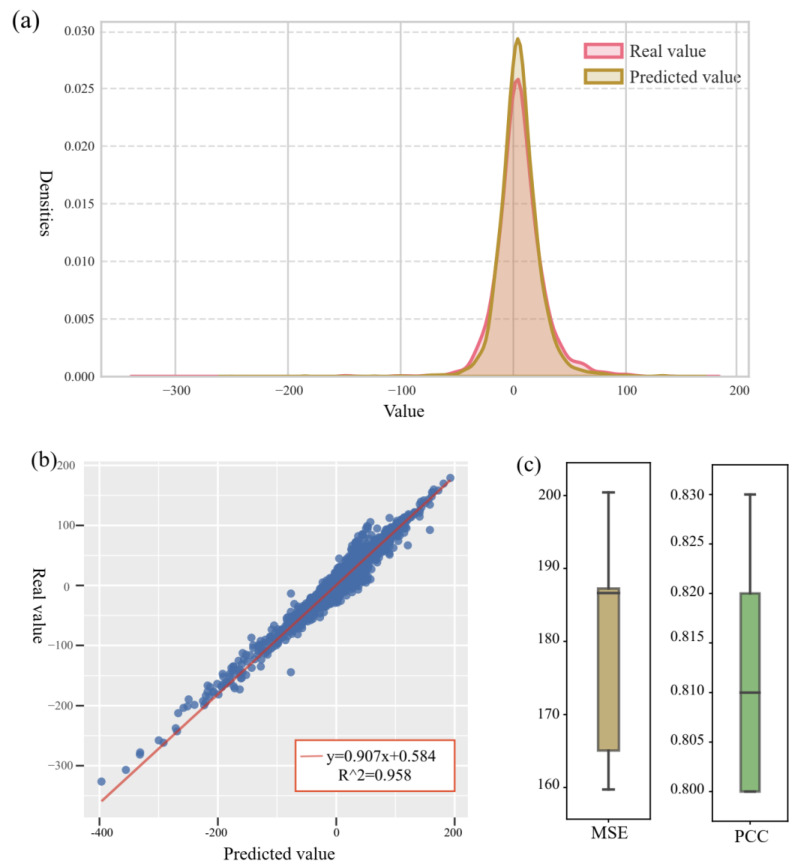
Comparison between our model’s prediction results and actual values. (**a**) shows the density plot of predicted and actual scores. (**b**) shows the correlation graph between the predicted and actual scores. (**c**) shows the experimental results of five-fold cross-validation based on the MMFSyn model using a box plot.

**Figure 5 biomolecules-14-01039-f005:**
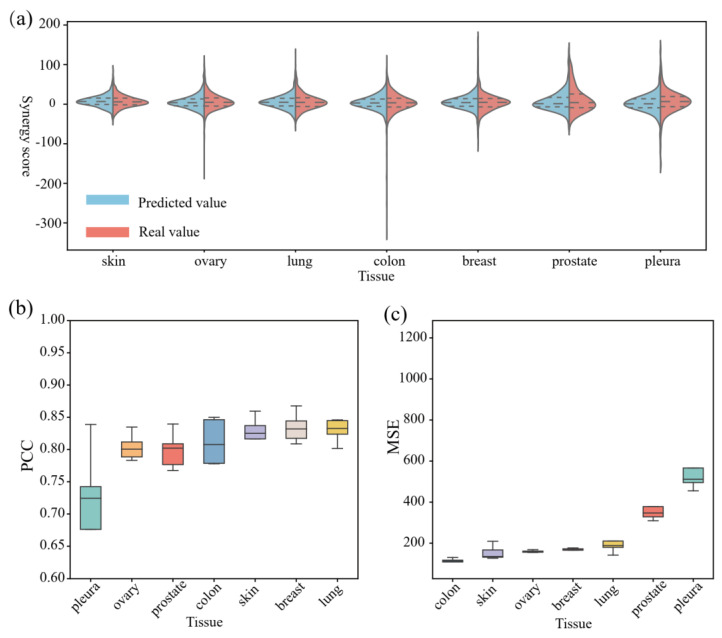
Comparison between our predicted results and actual values differentiated by organizational category. (**a**) shows a double-sided violin plot displaying the distribution of true collaborative scores of cell lines derived from each tissue and the distribution of predicted collaborative scores by the model. (**b**) shows a box plot of PCC values classified by organizational type. (**c**) shows a box plot of MSE values classified by organizational type.

**Figure 6 biomolecules-14-01039-f006:**
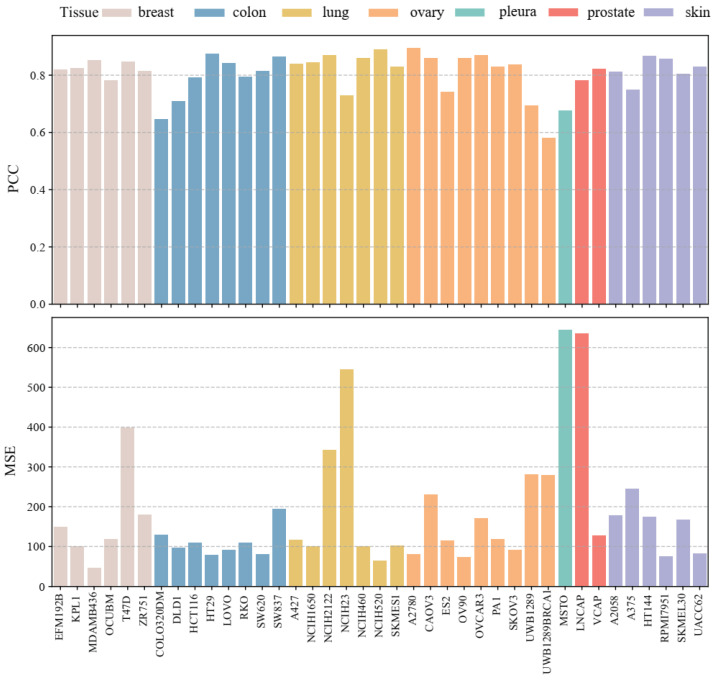
Predictive results by cell lines in different tissues.

**Figure 7 biomolecules-14-01039-f007:**
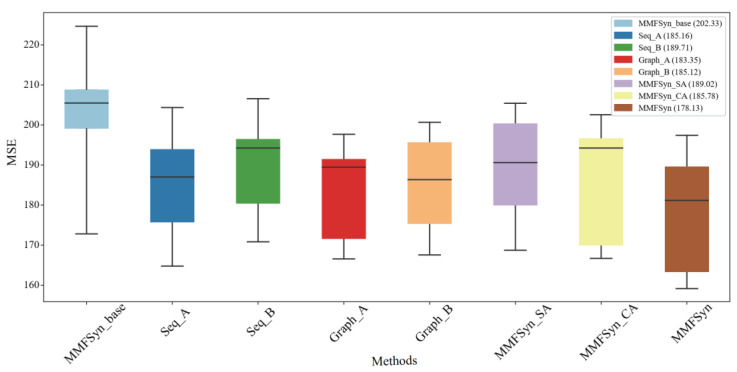
The 5-fold cross-validation results of different variants of MMFSyn on the dataset.

**Table 1 biomolecules-14-01039-t001:** Comparison results of the ablation study for cell lines.

Type	Method	MSE
DL	Cell_base	214.36 ± 44.83
DL	Cell_mut	189.69 ± 44.15
DL	Cell-gen	185.27 ± 42.93
DL	MMFSyn	178.13 ± 41.58

**Table 2 biomolecules-14-01039-t002:** Comparison results of different prediction methods.

Type	Method	MSE	RMSE	PCC	SCC
DL	PRODeepSyn	208.49 ± 42.53	14.54 ± 1.38	0.75 ± 0.02	0.74 ± 0.03
DL	TranSynergy	221.21 ± 41.15	14.88 ± 1.41	0.75 ± 0.02	0.73 ± 0.02
DL	AudnnSynergy	239.12 ± 45.55	15.46 ± 1.42	0.74 ± 0.03	0.72 ± 0.02
DL	DeepSynergy	245.49 ± 43.85	15.65 ± 1.56	0.71 ± 0.02	0.69 ± 0.02
ML	Linear Regression	480.46 ± 53.37	21.68 ± 1.36	0.47 ± 0.01	0.46 ± 0.02
ML	Elastic Net	415.36 ± 51.59	20.38 ± 1.35	0.46 ± 0.02	0.45 ± 0.03
ML	Random Forest	280.72 ± 42.37	16.65 ± 1.14	0.64 ± 0.03	0.63 ± 0.02
ML	XGBoost	296.34 ± 46.37	17.05 ± 1.32	0.66 ± 0.02	0.65 ± 0.03
DL	MMFSyn	**178.13 ± 41.58**	**13.33 ± 1.15**	**0.81 ± 0.02**	**0.80 ± 0.02**

**Table 3 biomolecules-14-01039-t003:** Comparison results of different prediction methods on an independent test set.

Type	Method	MSE
DL	PRODeepSyn	276.79 ± 49.94
DL	TranSynergy	289.69 ± 45.36
ML	XGBoost	396.54 ± 65.37
DL	MMFSyn	248.74 ± 46.93

## Data Availability

https://github.com/pluto-yt/MMFSyn (accessed on 7 August 2024).

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
