# Peer review of "MMFSyn: A Multimodal Deep Learning Model for Predicting Anticancer Synergistic Drug Combination Effect"

_biomolecules, 2024, doi:10.3390/biom14081039_

Round 1

Reviewer 1 Report

Comments and Suggestions for Authors

The article presents MMFSyn, a new model for predicting the synergy of anticancer drug combinations. The multimodal approach, which integrates different structural features of drugs and information on tumour cell lines, is promising and the model architecture appears well designed. However, the article presents several methodological criticalities that limit the interpretability and robustness of the results:

1. Size of the training set: The number of samples used to train the model is not clearly specified. Clearly report the number of molecules, cell lines and drug combinations used for training. If possible, indicate in the cited github the traning used including the various descriptors, to ensure the reproducibility of the method. 

2. Validation strategy: As far as I understand, the article relies solely on 5-fold cross-validation to assess the performance of the model. Although useful, this approach does not guarantee the model's ability to generalise to completely new data. It would be essential to complement the cross-validation with an external validation, using an independent dataset never seen during training.

3. Comparison with other models: It is unclear whether the comparison models were trained specifically for task. For example by taking an XGB and training it from scratch, or whether they are models already in the state of the art for the same objective applied to the dataset you built.

In conclusion, in my opinion while this is a promising approach the work can be greatly improved by specifying these three points.

Reviewer 2 Report

Comments and Suggestions for Authors

In the present study, the authors of article "MMFSyn: a multimodal deep learning model for predicting anticancer synergistic drug combination effect", suggested a model named MMFSyn, which shows promise.

I would like to read further revision of the current article to overcome following limitations:

1. I suppose the study relies on a single dataset from O'Neil et al. with 23,052 drug combination samples across 39 cancer cell lines. Using only one dataset limits the generalizability of the results. Validating on additional independent datasets would strengthen the conclusions. Also, while the computational model shows good performance, there is no experimental validation of the predicted synergistic drug combinations. Wet-lab experiments to confirm some of the top predictions would significantly bolster the practical utility of the approach.

2. The authors note that synergy scores are concentrated near 0 in the training set, leading to conservative predictions for highly synergistic combinations. This imbalance could bias the model and limit its ability to identify novel, highly synergistic pairs. Also, the model uses a large number of features and complex neural network architectures. There is a risk of overfitting to the training data, especially given the limited dataset size. More rigorous cross-validation or external validation would help address this concern.

3. Although the model achieves good predictive performance, its interpretability is limited. The complex deep learning architecture makes it challenging to understand which specific molecular or cellular features are driving the synergy predictions. Moreover, the complex multi-modal deep learning architecture likely requires significant computational resources to train and run. The practicality of using this model for large-scale virtual screening is not discussed.

4. While the authors compare to some other machine learning approaches, a more thorough comparison to simpler models (e.g. linear regression, random forests) would help justify the added complexity of the deep learning approach. Even if the model can predict synergy, it does not provide insights into the biological mechanisms underlying the synergistic effects. Incorporating pathway or network analysis could enhance the biological interpretability.

5. The dataset only includes 38 drugs. Testing on a more diverse set of compounds would better demonstrate the model's broad applicability. Also, using only 39 cancer cell lines may not capture the full diversity of tumor biology. Finally, if authors could test on primary patient samples or in vivo models would increase clinical relevance.

Reviewer 3 Report

Comments and Suggestions for Authors

This paper proposes a novel deep learning model MMFSyn based on multimodal drug data combined with cell line features. However, there are several critical issues which need to be addressed.

1.      There are many large public database (e.g. DrugComb, DrugCombDB, NCI-ALMANA, YNERGxDB, GDSC etc) providing drug synergy score, which has been popularly used in this domain. However, current study only use small data covering only 38 drugs from 39 cancer cell lines in 7 tissues with 583 drug combinations. It would be necessary to apply their method with existing large database and evaluate the performance.

2.      References on models developed to predict drug synergy or drug-drug interactions are lacking. For example, in line 57, additional reference can be added related with Drug synergy prediction as well as drug-drug interactions which is another way to identify the drug synergy.

                              i.           (DFFNDDS: prediction of synergistic drug combinations with dual feature fusion networks | Journal of Cheminformatics | Full Text (biomedcentral.com)

                             ii.           Stratification and prediction of drug synergy based on target functional similarity | npj Systems Biology and Applications (nature.com)

                           iii.           SYNDEEP: a deep learning approach for the prediction of cancer drugs synergy | Scientific Reports (nature.com)

                           iv.           Molecules | Free Full-Text | DEML: Drug Synergy and Interaction Prediction Using Ensemble-Based Multi-Task Learning (mdpi.com)

                             v.           Molecules | Free Full-Text | Prediction of Drug-Drug Interaction Using an Attention-Based Graph Neural Network on Drug Molecular Graphs (mdpi.com)

                           vi.           Pharmaceutics | Free Full-Text | PRID: Prediction Model Using RWR for Interactions between Drugs (mdpi.com)

                          vii.           Applied Sciences | Free Full-Text | DDI-SSL: Drug–Drug Interaction Prediction Based on Substructure Signature Learning (mdpi.com)

3.      There are some typos in the manuscript. So, the manuscript would be polished. For example,,in line 164, notation is mistyped. Also, the explanation of notation is not sufficient. All the term in the equation should be clearly defined.

4.      This study employs drug structures (e.g., SMILES) at various levels (e.g., 1D, 2D, 3D) as features. Given that these representations originate from the same source (e.g., drug SMILES) but in different forms, there is a possibility of redundancy. It is essential to verify whether these features are redundant, for instance, through a correlation matrix. Alternatively, the authors should evaluate the performance using each feature individually to determine the extent of performance improvement.

5.      How was tissue information handled? In cancer data related to drug synergy or sensitivity, tissue type information is as crucial as cell line information. This needs to be explained.

6.      In the current model, the same expression data and mutation data matrix are always used regardless of the drug pairs. As a result, these datasets might not effectively differentiate the synergy between drug pairs. In other words, the structure information (e.g. 1d, 2d, 3d) alone may be determining drug pair synergy. It is essential to quantify the contribution of expression and mutation data to the prediction of drug synergy. Demonstrating the impact of expression/mutation data on identifying drug synergy is critical. The authors should show whether the ranking of drug pairs based on the synergy score changes when expression/mutation data are included in the analysis

Round 2

Reviewer 2 Report

Comments and Suggestions for Authors

All comments were addressed.

Reviewer 3 Report

Comments and Suggestions for Authors

The authors have thoroughly addressed all issues. Thank you.